# VARIATIONAL QUANTUM ADABOOST WITH SUPERVISED LEARNING GUARANTEE

## ABSTRACT

Although variational quantum algorithms based on parameterized quantum circuits promise to achieve quantum advantages, in the noisy intermediate-scale quantum (NISQ) era, their capabilities are greatly constrained due to limited number of qubits and depth of quantum circuits. Therefore, we may view these variational quantum algorithms as weak learners in supervised learning. Ensemble methods are a general technique in machine learning for combining weak learners to construct a more accurate one. In this paper, we theoretically prove and numerically verify a learning guarantee for variational quantum adaptive boosting (AdaBoost). To be specific, we theoretically depict how the prediction error of variational quantum AdaBoost on binary classification decreases with the increase of the number of boosting rounds and sample size. By employing quantum convolutional neural networks, we further demonstrate that variational quantum AdaBoost can not only achieve much higher accuracy in prediction, but also help mitigate the impact of noise. Our work indicates that in the current NISQ era, introducing appropriate ensemble methods is particularly valuable in improving the performance of quantum machine learning algorithms.

## 1 INTRODUCTION

### 1.1 BACKGROUND

Machine learning has achieved remarkable success in various fields with a wide range of applications (Mohri et al., 2018; Jordan & Mitchell, 2015; Butler et al., 2018; Genty et al., 2021). A major objective of machine learning is to develop efficient and accurate prediction algorithms, even for large-scale problems (Zhang et al., 2022; Ergun et al., 2022; Lyle et al., 2022). The figure of merit, prediction error, can be decomposed into the summation of training and generalization errors. Both of them should be made small to guarantee an accurate prediction. However, there is a tradeoff between reducing the training error and restricting the generalization error through controlling the size of the hypothesis set, known as Occam's Razor principle (Rasmussen & Ghahramani, 2000; Mohri et al., 2018).

For classical machine learning, empirical studies have demonstrated that the training error can often be effectively minimized despite the non-convex nature and abundance of spurious minima in training loss landscapes (Livni et al., 2014; Du et al., 2019; Arora et al., 2018). This observation has been explained by the theory of over-parameterization (Jacot et al., 2018; Nitanda & Suzuki, 2021; Zhang et al., 2017; Arora et al., 2020; 2019; Oymak & Soltanolkotabi, 2020). However, it is still difficult to theoretically describe how to guarantee a good generalization, which is one of the key problems to be solved in classical machine learning.

Owing to the immense potential of quantum computing, extensive efforts have been dedicated to developing quantum machine learning (Biamonte et al., 2017; Carleo & Troyer, 2017; Dunjko & Briegel, 2018; Carleo et al., 2019; Cerezo et al., 2022; Qi et al., 2023). However, in the noisy intermediate-scale quantum (NISQ) era, the capability of quantum machine learning is greatly constrained due to limited number of qubits and depth of the involved quantum circuits. Algorithms based on parameterized quantum circuits (PQCs) have become the leading candidates to yield potential quantum advantages in the era of NISQ (Landman et al., 2023; Jerbi et al., 2021; Du et al.,

2020). The basic idea behind them is that these parameterized quantum models can provide representational and/or computational powers beyond what is possible with classical models (Schuld et al., 2021; Liu et al., 2021; Huang et al., 2021). There are mainly three kinds of parameterized quantum models (Jerbi et al., 2023): (a) explicit models (Cerezo et al., 2021a; Benedetti et al., 2019), where data are first encoded into quantum states, after undergoing a PQC, the quantum states are measured and the information is used to update the variational parameters through a classical routine; (b) implicit kernel models (Havlíček et al., 2019; Schuld & Killoran, 2019), where the kernel matrices of the encoding data are computed through quantum circuits, and then used to label data; (c) re-uploading models (Pérez-Salinas et al., 2020), where encoding and parameterized circuits are interleaved. A unified framework has been set in Jerbi et al. (2023) for the three quantum models, and it was pointed out that the advantages of quantum machine learning may lie beyond kernel methods. They found that although kernel methods are guaranteed to achieve a lower training error, their generalization power is poor. Thus, both the training and generalization errors should be taken into account when evaluating the prediction accuracy.

It has been proved in Caro et al. (2022) that good generalization can be guaranteed from few training data for a wide range of quantum machine learning models. However, in contrast to the classical case, training quantum models is notoriously difficult as it often suffers from the phenomena of barren plateaus (McClean et al., 2018; Haug et al., 2021; Cerezo et al., 2021b; Ortiz Marrero et al., 2021; Wang et al., 2021a; Zhao & Gao, 2021), where the cost gradient vanishes exponentially fast, and there exist (exponentially) many spurious local minima (Anschuetz, 2022; Anschuetz & Kiani, 2022; You & Wu, 2021). In this sense, most quantum learning algorithms can be viewed as weak learners in the language of supervised machine learning.

To improve the performance of quantum algorithms, we can employ ensemble methods as inspired by the classical ensemble learning. There are various kinds of ensemble methods, e.g., bagging (Breiman, 1996), plurality voting (Lam & Suen, 1997; Lin et al., 2003) and boosting (Freund et al., 1999). It has been suggested in Jiang et al. (2020) that an optimized weighted mixture of concepts, e.g., PAC-Bayesian (McAllester, 1999), is a promising candidate for further research. Thus, adaptive boosting (AdaBoost), which adaptively adjusts the weights of a set of base learners to construct a more accurate learner than base learners, is appropriate for improving the performance of quantum weak learners. For classical machine learning, there has been a rich theoretical analysis on AdaBoost (Freund & Schapire, 1997; Bartlett et al., 1998; Mohri et al., 2018; Grønlund et al., 2019), and it has been shown to be effective in practice (Sun et al., 2021; Drucker et al., 1993; Li et al., 2008; Zhang et al., 2019). In this paper, we provide the first theoretical learning guarantee for binary classification of variational quantum AdaBoost, and then numerically investigate its performance on 4-class classification by employing quantum convolutional neural networks (QCNNs), which are naturally shallow and particularly useful in NISQ era.

## 1.2 RELATED WORK

Various quantum versions of classical AdaBoost have been proposed, such as Arunachalam & Maity (2020); Wang et al. (2021b); Ohno (2022). In their works, they employed quantum subroutines, e.g., mean estimation and amplitude amplification, to update quantum weak classifiers and estimate the weighted errors to reduce the time complexity. Therefore, the realizations of these quantum versions of AdaBoost are beyond the scope of current NISQ circuits. In contrast, in this work we utilize variational quantum classifiers realized on the current NISQ circuits, which are obtained through a quantum-classical hybrid way.

Recently, ensemble methods have been proposed to enhance the accuracy and robustness of quantum classification with NISQ devices. Variational quantum AdaBoost and variational quantum Bagging have been empirically investigated in Li et al. (2023); Incudini et al. (2023) with hardware-efficient ansatz. It was demonstrated via simulations that quantum AdaBoost not only outperforms quantum Bagging (Li et al., 2023), but also can save resources in terms of the number of qubits, gates, and training samples (Incudini et al., 2023).

## 1.3 OUR CONTRIBUTIONS

In this paper, we theoretically and numerically investigate the performance of variational quantum AdaBoost by focusing on classification. Our contributions are summarized as follows.

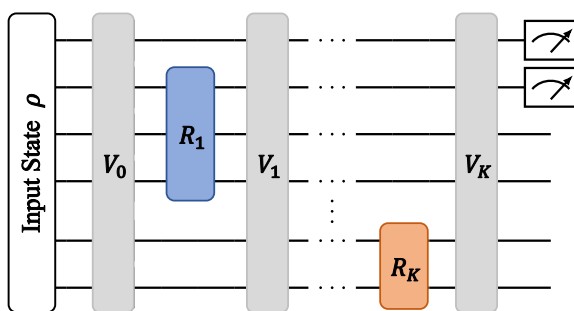

Figure 1: The schematic of a PQC with $K$ independent trainable gates. Each trainable gate is parameterized by a multi-qubit rotational gate which is efficiently implementable.

1) For binary classification, we provide the first theoretical upper bound on the prediction error of variational quantum AdaBoost, demonstrating how the prediction error converges to 0 as the increase of the number of boosting rounds and sample size.

2) We numerically demonstrate that variational quantum AdaBoost can achieve a higher level of prediction accuracy as compared to quantum Bagging, classical AdaBoost and classical Bagging. We further demonstrate that with only few boosting rounds variational quantum AdaBoost can help mitigate the impact of noises and achieve better performance than noiseless models, which is particularly valuable for potential applications, especially in the NISQ era.

The paper is organized as follows. In Section 2, we briefly introduce the quantum classifier and variational quantum AdaBoost. In Section 3, we present our theoretical and empirical results on the performance of variational quantum AdaBoost. Section 4 concludes the paper.

## 2 QUANTUM CLASSIFIER AND ADABOOST

### 2.1 QUANTUM CLASSIFIER

We start with briefly introducing some quantum notation. In quantum computing, information is described in terms of quantum states. For an $N$-qubit system, the quantum state $\rho$ can be mathematically represented as a positive semi-definite Hermitian matrix $\rho \in \mathbb{C}^{2^N \times 2^N}$ with $\text{Tr}[\rho] = 1$. The elementary quantum gates are mathematically described by unitary matrices. A quantum gate $U$ acting on a quantum state $\rho$ takes the state to the output state as $U\rho U^\dagger$ where $U^\dagger$ is the conjugate and transpose of $U$. When measuring an observable $O$ (a Hermitian operator) at quantum state $\rho$, its expectation is $\text{Tr}[O\rho]$.

For a $D$-class classification problem, suppose that both the training and test data are independent and identically distributed (i.i.d.) according to some fixed but unknown distribution $\mathcal{D}$ defined over the sample and label space $\mathcal{X} \times \mathcal{Y}$. When the sample set $S = \{(\boldsymbol{x}_i, y_i)\}_{i=1}^n$ are classical, we can first choose a quantum encoding circuit to embed the classical data $\boldsymbol{x}_i$ into quantum state $\rho(\boldsymbol{x}_i)$ (Lloyd et al., 2020; Schuld et al., 2021; Goto et al., 2021), which is the explicit quantum model under consideration. Without loss of generality, we only consider the case where the data are quantum in the following, namely, $S = \{(\rho(\boldsymbol{x}_i), y_i)\}_{i=1}^n \subset \mathcal{X} \times \mathcal{Y}$. For a $D$-class classification, $\mathcal{Y} = \{1, \cdots, D\} \triangleq [D]$.

To label $\rho(\boldsymbol{x})$, a quantum hypothesis or classifier $h_{\boldsymbol{\theta}}(\cdot)$ can be described in the form of

$$h_{\boldsymbol{\theta}}(\boldsymbol{x}) = \arg\max_{d \in [D]} \text{Tr}\left[P_d U(\boldsymbol{\theta}) \rho(\boldsymbol{x}) U^\dagger(\boldsymbol{\theta})\right]. \tag{1}$$

Here, $\{P_d\}_{d=1}^D$ are disjoint projectors with $P_d$ relating to the $d$-th class for $d \in [D]$, and $U(\boldsymbol{\theta})$ describes the action of a PQC with $\boldsymbol{\theta}$ being the trainable or variational parameters.

To be specific, as illustrated in Fig. 1, suppose that the employed PQC is composed of a total number of $K$ independent parameterized gates and non-trainable gates $\{V_k\}_{k=0}^K$, whose action can be

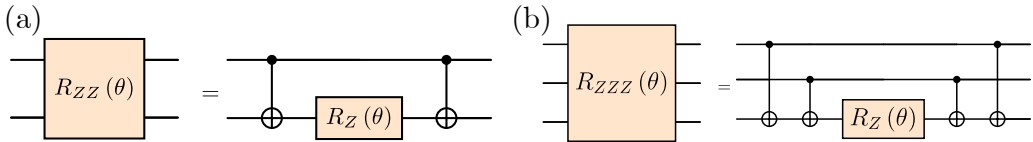

Figure 2: Hardware-efficient implementations of multi-qubit rotational gates. (a) The module of 2-qubit rotational gate $R_{ZZ}(\theta)$ around the Pauli operator $Z \otimes Z$. (b) The module of 3-qubit rotational gate $R_{ZZZ}(\theta)$ around the Pauli operator $Z \otimes Z \otimes Z$.

described as

$$U(\boldsymbol{\theta}) = \prod_{k=1}^{K} \left[ V_k R_k^{(i_k, j_k)}(\theta_k) \right] \cdot V_0, \tag{2}$$

where $\boldsymbol{\theta} = (\theta_1, \cdots, \theta_K)$ denotes a $K$-dimensional parameter vector. For each $k$, the trainable gate $R_k^{(i_k, j_k)}(\theta_k)$ denotes a rotational gate with angle $\theta_k$ around a $j_k$-qubit Pauli tensor product operator $P_k$, which acts non-trivially on the $i_k$-th through to $(i_k + j_k - 1)$-th qubits, namely,

$$\begin{aligned} R_k^{(i_k, j_k)}(\theta_k) =& I^{\otimes(i_k-1)} \otimes e^{-i\frac{\theta_k}{2} P_k} \otimes I^{\otimes(n-i_k-j_k+1)} \\ =& I^{\otimes(i_k-1)} \otimes \left( \cos\frac{\theta_k}{2} I^{\otimes j_k} - i \sin\frac{\theta_k}{2} P_k \right) \otimes I^{\otimes(n-i_k-j_k+1)}. \end{aligned}$$

In practice, these multi-qubit rotational gates can be implemented by a series of single-qubit gates and typical 2-qubit controlled gates, which are efficient to realize. For example, as illustrated in Fig. 2, the multi-qubit rotational gates around $z$ axis can be implemented by a single-qubit rotational gate around $z$ axis and some 2-qubit CNOT gates.

The prediction error or expected risk of the quantum hypothesis function $h_{\boldsymbol{\theta}}$ is defined as

$$R(h_{\boldsymbol{\theta}}) = \mathop{\mathbb{E}}_{(\boldsymbol{x},y)\sim\mathcal{D}} \mathbb{I}_{h_{\boldsymbol{\theta}}(\boldsymbol{x})\neq y} = \mathop{\mathbb{P}}_{(\boldsymbol{x},y)\sim\mathcal{D}} [h_{\boldsymbol{\theta}}(\boldsymbol{x}) \neq y]. \tag{3}$$

The prediction error of a hypothesis is not directly accessible, since both the label of unseen data and the distribution $\mathcal{D}$ are unavailable. However, we can take the training error or empirical risk of $h_{\boldsymbol{\theta}}$ as a proxy, defined as

$$\widehat{R}_S(h_{\boldsymbol{\theta}}) = \frac{1}{n} \sum_{i=1}^{n} \mathbb{I}_{h_{\boldsymbol{\theta}}(\boldsymbol{x}_i)\neq y_i}. \tag{4}$$

The difference between the prediction error $R(h_{\boldsymbol{\theta}})$ and the training error $\widehat{R}_S(h_{\boldsymbol{\theta}})$ is referred to as the generalization error, which reads

$$\mathtt{gen}(h_{\boldsymbol{\theta}}) = R(h_{\boldsymbol{\theta}}) - \widehat{R}_S(h_{\boldsymbol{\theta}}). \tag{5}$$

It is clear that to make accurate predictions, both the training and generalization errors should be small.

## 2.2 Variational Quantum AdaBoost

We denote by $\mathcal{H}$ the hypothesis set which is composed of base classifiers $h_{\boldsymbol{\theta}}(\cdot)$ in the form of Eq. (1). Inspired by classical multi-class AdaBoost (Hastie et al., 2009), the procedure of variational quantum AdaBoost is presented in Algorithm 1, which is similar to that in Li et al. (2023).

Algorithm 1 has the input including a labeled sample set $S = \{(\rho(\boldsymbol{x}_i), y_i)\}_{i=1}^{n}$, the number of boosting rounds $T$ typically selected via cross-validation, and maintains a distribution over the indices $[n]$ for each round. The initial distribution is assumed to be uniform, i.e., $\mathcal{D}_1(i) = \frac{1}{n}$. At each round of boosting, i.e., for each $t \in [T]$, given a classifier $h_t \in \mathcal{H}$, its error $\epsilon_t$ on the training data weighted by the distribution $\mathcal{D}_t$ reads

$$\epsilon_t = \sum_{i=1}^{n} \mathcal{D}_t(i) \mathbb{I}_{h_t(\boldsymbol{x}_i)\neq y_i}. \tag{6}$$

---

**Algorithm 1:** $D$-Class Variational Quantum AdaBoost

---

**input:** Hypothesis set $\mathcal{H} = \{h_{\boldsymbol{\theta}}\}$
    Sample set $S = \{(\rho(\boldsymbol{x}_i), y_i)\}_{i=1}^n$
    Boosting rounds $T$
    Distribution $\mathcal{D}_1(i) = \frac{1}{n}$, for $i \in [n]$

**for** $t \leftarrow 1$ **to** $T$ **do**
  $h_t \leftarrow$ base classifier in $\mathcal{H}$ with error $\epsilon_t < \frac{D-1}{D}$

  $\alpha_t \leftarrow \log \frac{1-\epsilon_t}{\epsilon_t} + \log(D-1)$

  **for** $i \leftarrow 1$ **to** $n$ **do**
   $\big|$   $\mathcal{D}_{t+1}(i) \leftarrow \mathcal{D}_t(i) \exp\left[\alpha_t \mathbb{I}_{y_i \neq h_t(\boldsymbol{x}_i)}\right]$
  **end**

  normalize $\{\mathcal{D}_{t+1}(i)\}_{i=1}^n$
**end**

$f \leftarrow \underset{d \in [D]}{\arg\max} \sum_{t=1}^T \alpha_t \mathbb{I}_{h_t = d}$

**output:** Predictor $f$

---

We choose a weak classifier $h_t$ such that $\epsilon_t < \frac{D-1}{D}$, which is easily satisfied. Then the distribution is updated as $\mathcal{D}_{t+1}(i) \propto \mathcal{D}_t(i) \exp\left[\alpha_t \mathbb{I}_{y_i \neq h_t(\boldsymbol{x}_i)}\right]$, where $\alpha_t = \log \frac{1-\epsilon_t}{\epsilon_t} + \log(D-1)$. After $T$ rounds of boosting, Algorithm 1 returns the $D$-class quantum AdaBoost classifier.

## 3   MAIN RESULTS

### 3.1   BINARY VARIATIONAL QUANTUM ADABOOST GUARANTEE

For multi-class classification, an alternative approach is to reduce the problem to that of multiple binary classification tasks. For each task, a binary classifier is returned, and the multi-class classifier is defined by a combination of these binary classifiers. Two standard reduction techniques are one-versus-the-rest and one-versus-one (Aly, 2005; Mohri et al., 2018). In this subsection, we focus on the basic binary variaitonal quantum AdaBoost, and theoretically establish its learning guarantee.

For binary classification, it is more convenient to denote the label space by $\mathcal{Y} = \{-1, +1\}$. The base quantum hypothesis $h_{\boldsymbol{\theta}}(\cdot)$ can be defined in terms of the Pauli-$Z$ operator $Z = \begin{pmatrix} 1 & 0 \\ 0 & -1 \end{pmatrix}$ as

$$h_{\boldsymbol{\theta}}(\boldsymbol{x}) = \text{Tr}\left[ Z U(\boldsymbol{\theta}) \rho(\boldsymbol{x}) U^\dagger(\boldsymbol{\theta}) \right], \tag{7}$$

whose range is $[-1, +1]$, and its sign is used to determine the label, namely, we label $-1$ when $h_{\boldsymbol{\theta}}(\boldsymbol{x}) \leq 0$; otherwise, it is labeled as $+1$.

It is straightforward to verify that for a sample $(\rho(\boldsymbol{x}), y)$, the following important relation holds:

$$\mathbb{I}_{y \neq h_{\boldsymbol{\theta}}(\boldsymbol{x})} = \mathbb{I}_{y h_{\boldsymbol{\theta}}(\boldsymbol{x}) \leq 0}. \tag{8}$$

By employing Eq. (8) and inspired by the classical binary AdaBoost (Mohri et al., 2018), we can modify Algorithm 1 slightly to make it more suitable for binary classification as presented in Algorithm 2, and further establish the learning guarantee for the binary variational quantum AdaBoost.

Different from Algorithm 1, the hypothesis set $\mathcal{H}$ in Algorithm 2 is composed of quantum hypothesis in the form of Eq. (7). At each round of boosting, a new classifier $h_t \in \mathcal{H}$ is selected such that its error $\epsilon_t < \frac{1}{2}$, and the distribution update role

$$\mathcal{D}_{t+1}(i) = \frac{\mathcal{D}_t(i) \exp\left(-\alpha_t y_i h_t(x_i)\right)}{Z_t}$$

is different from what is used in Algorithm 1. It can be verified that $\alpha_t = \frac{1}{2} \log \frac{1-\epsilon_t}{\epsilon_t}$ is chosen to minimize the upper bound of the empirical risk $\widehat{R}_S(f)$ of the binary variational quantum AdaBoost (Mohri et al., 2018).

---

**Algorithm 2:** Binary Variational Quantum AdaBoost

---

**input:** Hypothesis set $\mathcal{H} = \{h_{\boldsymbol{\theta}}\}$
   Sample set $S = \{(\rho(\boldsymbol{x}_i), y_i)\}_{i=1}^{n}$
   Boosting rounds $T$
   Distribution $\mathcal{D}_1(i) = \frac{1}{n}$, for $i \in [n]$

**for** $t \leftarrow 1$ **to** $T$ **do**
  $h_t \leftarrow$ base classifier in $\mathcal{H}$ with ~~small~~ error $\epsilon_t < \frac{1}{2}$

  $\alpha_t \leftarrow \frac{1}{2} \log \frac{1-\epsilon_t}{\epsilon_t}$

  $Z_t \leftarrow 2[\epsilon_t(1-\epsilon_t)]^{\frac{1}{2}}$   % normalization factor

  **for** $i \leftarrow 1$ **to** $n$ **do**
    $\mathcal{D}_{t+1}(i) \leftarrow \frac{\mathcal{D}_t(i) \exp[-\alpha_t y_i h_t(x_i)]}{Z_t}$
  **end**
**end**
$f \leftarrow \operatorname{sgn}\left(\sum_{t=1}^{T} \alpha_t h_t\right)$

**output:** Predictor $f$

---

The performance of the binary variational quantum AdaBoost is guaranteed by the following theorem, whose proof can be found in Appendix B.

**Theorem 3.1.** *For the binary variational quantum AdaBoost Algorithm 2, assume that there exists $\gamma > 0$ such that $\epsilon_t \leq \frac{1}{2} - \gamma$, for each $t \in [T]$, and the employed PQC has a total number of $K$ independent parameterized gates. Then for any $\delta > 0$, with a probability at least $1 - \delta$ over the draw of an i.i.d. $n$-size sample set, the prediction error $R(f)$ of the returned binary variational quantum AdaBoost classifier $f$ satisfies*

$$R(f) \leq e^{-2\gamma^2 T} + 12\sqrt{\frac{K \log 7K}{n}} + 4\sqrt{\frac{K}{n}} + \sqrt{\frac{\log \frac{1}{\delta}}{2n}}. \tag{9}$$

It is clear that Theorem 3.1 provides a solid and explicit learning guarantee for binary variational quantum AdaBoost classifier. The first term in the RHS of Eq. (9) describes the upper bound of the empirical error $\widehat{R}_S(f)$, which decreases exponentially fast as a function of the boosting rounds $T$ owing to the good nature of AdaBoost. The last three terms describe the upper bound of the generalization error $\texttt{gen}(f)$. Here, in contrast to the classical case, our success in bounding the generalization error owes to the good generalization property of quantum machine learning. As the number of independent trainable gates $K$ increases, the hypothesis set $\mathcal{H}$ becomes richer. Thus, the second and third terms in the RHS of Eq. (9) depict the penalty of the complexity of the hypothesis set $\mathcal{H}$ on the generalization.

In the NISQ era, it is important to take into account of the effect of noise originating from various kinds of sources. From the detailed proof of Theorem 3.1, it can be verified that for noisy PQCs, as long as there is an edge $\gamma > 0$ between our base classifiers and the completely random classifier, that is $\epsilon_t < \frac{1}{2} - \gamma$ for all $t \in [T]$, the learning performance of the variational quantum AdaBoost can also be guaranteed. However, it is worth pointing out that this edge assumption will become hard to be met when there is very large noise.

## 3.2   Numerical Experiments for 4-Class Classification

In this subsection, we numerically investigate the performance of 4-class variational quantum AdaBoost. To be specific, our task is to perform a 4-class classification of the handwritten digits $\{0, 1, 2, 3\}$ in MNIST datasets (LeCun et al., 1998). In our numerical experiments, we employ QCNN as our PQC, which has been proven free of barren plateau (Pesah et al., 2021) and has been widely used as quantum classifiers (Wei et al., 2022; Chen et al., 2022; Hur et al., 2022).

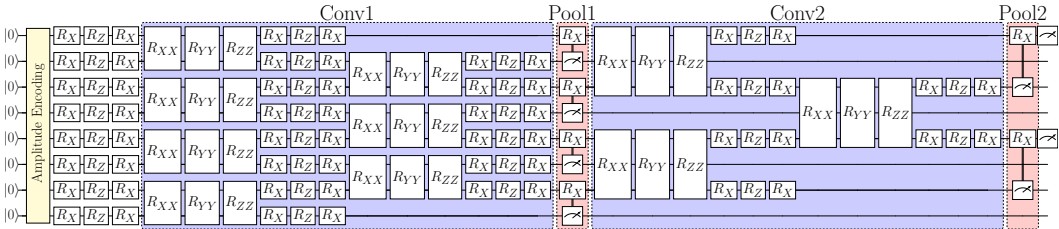

Figure 3: The architecture of QCNN. After amplitude encoding, a set of universal rotational gates are applied to each qubit, followed by two blocks of convolutional (Conv) and pooling (Pool) layers. The pooling layers not only reduce the system size, but also provide non-linearity for the whole circuit.

For the $D$-class variational quantum AdaBoost algorithm (here $D = 4$), at each round $t \in [T]$ where $T > 1$, to find a base classifier $h_t$ such that its error $\epsilon_t < \frac{D-1}{D}$, we need to optimize the variational parameters in QCNN. To do this, we optimize the following weighted cross-entropy loss function:

$$\min_{\boldsymbol{\theta}} \mathcal{L}\left(\boldsymbol{\theta}; S\right) = -\sum_{i=1}^{n} \mathcal{D}_t\left(i\right) \boldsymbol{y}_i^{\top} \log\left(\boldsymbol{p}_i\right), \tag{10}$$

where each label $y_i \in [D]$ has been transformed into a $D$-dimensional one-hot vector denoted by $\boldsymbol{y}_i$, and $\boldsymbol{p}_i = \left[p_{i,1}, \cdots, p_{i,D}\right]^{\top}$ with

$$p_{i,d} = \text{Tr}\left[P_d U\left(\boldsymbol{\theta}\right) \rho\left(\boldsymbol{x}_i\right) U^{\dagger}\left(\boldsymbol{\theta}\right)\right]$$

for each $d \in [D]$.

We employ Adam (Kingma & Ba, 2015) with learning rate $0.05$ as the optimizer and compute the loss gradient using the parameter-shift-rule (Romero et al., 2018; Mitarai et al., 2018; Schuld et al., 2019). We initialize the parameters of PQC according to standard normal distribution and stop optimizing when reaching the maximum number of iterations, which is set as $120$. The base classifier having the minimum training error $\epsilon_t$ in $120$ iterations is returned as $h_t$, whose error always satisfies $\epsilon_t < \frac{D-1}{D}$ in our experiments. When illustrating our results, like most of practical supervised learning tasks, we adopt the *accuracy* as the figure of merit, which is simply equal to 1 minus error.

In our first experiment, we employ a noiseless 8-qubit QCNN as the base classifier as illustrated in Fig. 3. We randomly sample two different 8000-size sample sets for training and testing, respectively. For each sampled image in MNIST, we first downsample it from $28 \times 28$ to $16 \times 16$ and then embed it into the QCNN using amplitude encoding. We conduct five experiments in total. Since the results of the five experiments are similar, to clearly demonstrate the difference between the training and test accuracy, we only randomly select one experiment and illustrate the results in Fig. 4. To demonstrate the positive effect of boosting operations, we also consider a classifier without any boosting which is referred to as QCNN-best. For QCNN-best, we optimize the QCNN for at most 3000 iterations, the same number as that in variational quantum Adaboost with $T = 25$ boosting rounds, and return the classifier having the best training accuracy. The prediction accuracy of QCNN-best is illustrated in Fig. 4 by the black dotted line. Without boosting QCNN-best can only achieve a prediction accuracy of $0.87$. It is clear that quantum Adaboost outperforms QCNN-best only after 3 rounds of boosting, and its performance can exceed $0.97$ after $T > 20$ rounds of boosting. Thus, to improve the prediction accuracy, boosting is much better than simply increasing the number of optimizations. Moreover, variational quantum AdaBoost maintains a good generalization throughout the entire training process as the differences between the training and prediction accuracy are always below $0.01$.

We further compare our variational quantum AdaBoost (QCNN+AdaBoost) with three other ensemble methods. The first one is variational quantum Bagging (QCNN+Bagging), the second is classical neural networks (CNN) with AdaBoost, referred to as CNN+AdaBoost, and the third one is CNN powered by Bagging, abbreviated as CNN+Bagging. The CNN takes the form of $f(x) = \sigma(W_2\sigma(W_1x+b_1)+b_2)$, where $\sigma(\cdot)$ denotes the softmax function and $W_1 \in \mathbb{R}^{3 \times 256}, W_2 \in$

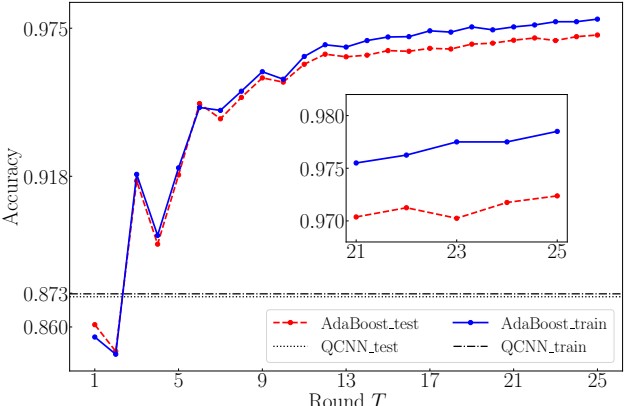

Figure 4: Accuracy of 4-class classification of variational quantum AdaBoost and QCNN-best in the noiseless case. The blue solid (red dashed) line depicts the training (testing) accuracy of variational quantum AdaBoost versus the boosting round $T$. The black dash-dotted (dotted) depicts the training (testing) accuracy of QCNN-best. It is clear that variational quantum AdaBoost can achieve a higher level of prediction accuracy (exceeding $0.97$ when the boosting round $T > 20$). During the whole process, the differences between the training and testing accuracy of AdaBoost are always below $0.01$, which indicates a good generalization of variational quantum AdaBoost.

$\mathbb{R}^{4\times 3}$, $b_1 \in \mathbb{R}^3$, $b_2 \in \mathbb{R}^4$. For Bagging methods, each base classifier is trained on a subset obtained by resampling the original training dataset for 8000 times, and the predictions of base classifiers are integrated through voting (Breiman, 1996). For the four ensemble methods to be compared, we utilize the same experimental setting. Specifically, the learning rate is set to be 0.05 and all the parameters are initialized according to standard normal distribution. We select the classifier having the smallest training error over 120 optimization iterations as the base classifier. The final strong classifier is chosen to be the one with the best training accuracy among the rounds from 1 to 25. We perform each ensemble method for five times, and demonstrate the results in Table 1. Note that there are 120 parameters in QCNN, while the number of paramters in CNN is 787. We find that although having more parameters, the training accuracy of classical ensemble methods is higher than their quantum counterparts. However, owing to the quantum advantage in generalization, our variational quantum AdaBoost (QCNN+AdaBoost) has the best prediction accuracy among the four ensemble methods. Although the training accuracy of QCNN+Bagging is poor, its generalization error is smaller than those of the classical ensemble methods. This is also attributed to the quantum advantage in generalization.

Table 1: Comparison between four different ensemble methods. The first row represents the training accuracy (acc.), the second row represents the prediction accuracy, and the third row describes the prediction accuracy of the first base classifier for different ensemble methods. The values in the table represent the mean values $\pm$ standard deviation.

|  | QCNN+AdaBoost | QCNN+Bagging | CNN+AdaBoost | CNN+Bagging |
|---|---|---|---|---|
| Training Acc. | $0.975\pm0.002$ | $0.898\pm0.006$ | $0.980\pm0.004$ | $0.982\pm0.004$ |
| Prediction Acc. | $\mathbf{0.973\pm0.001}$ | $0.888\pm0.005$ | $0.967\pm0.003$ | $0.965\pm0.002$ |
| Base Classifier | $0.861\pm0.019$ | $0.851\pm0.020$ | $0.876\pm0.051$ | $0.872\pm0.045$ |

In addition, we investigate the performance of variational quantum AdaBoost in the presence of noise. In practice, single-qubit gates can be implemented with a high level of fidelity, while the fidelity of implementing 2-qubit gates remains relatively lower. To take into account of this effect, we simulate a noisy 6-qubit QCNN, and consider three typical classes of noises: depolarizing noise, amplitude damping noise, and phase damping noise. After each involved 2-qubit gate we add a noise channel with noise probability $p = 0.03$. We randomly sample two different 1000-size sample sets for training and testing, respectively. For each sampled image, we first downsample it from $28 \times 28$ to $8 \times 8$ and then use the amplitude encoding to embed it into the QCNN. We illustrate the prediction accuracy of variational quantum AdaBoost in Fig. 5. For comparison, we also consider another two

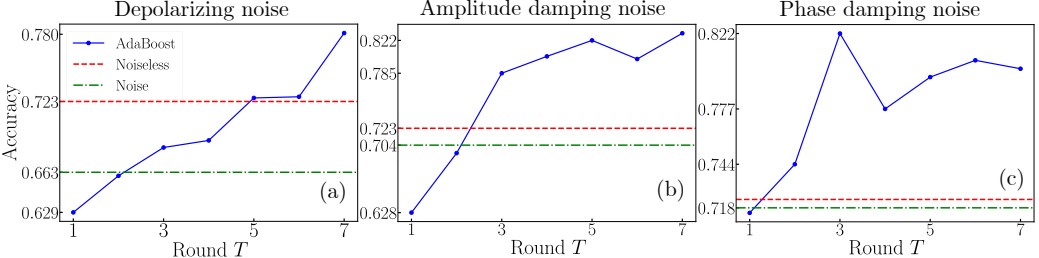

Figure 5: Prediction accuracy of variational quantum AdaBoost in the presence of noise. (a) QCNN with depolarizing noise. (b) QCNN with amplitude damping noise. (c) QCNN with phase damping noise. The blue solid line depicts the performance of variational quantum AdaBoost. The green dash-dotted line describes the prediction accuracy of the classifier employing the noisy QCNN, while the red dashed line depicts that with the ideally noiseless QCNN. Both of them do not take boosting operation. Variational quantum AdaBoost outperforms the noiseless classifier after at most 5 rounds of boosting implying that AdaBoost can help mitigate the impact of different kinds of noise.

classifiers having no boosting operations. One (red dashed) is returned by using an ideally noiseless QCNN, and the other (green dash-dotted) is obtained by employing the noisy QCNN. For both of them, we optimize the PQC for at most $840$ iterations, which is the same number as that in 7 rounds of variational quantum AdaBoost, and return the classifier having the best testing accuracy. The reason why their prediction accuracy is lower than that in Fig. 4 is that here we compress the images into $8 \times 8$ format, while the images in Fig. 4 are compressed into $16 \times 16$. Excessive compression leads to loss of information, thus reducing the overall prediction accuracy of the classifier. We find that for the three typical classes of noises, variational quantum AdaBoost outperforms the noiseless classifier after at most 5 rounds of boosting. This implies that AdaBoost can help mitigate the impact of different kinds of noises, which is particularly useful in the NISQ era. The reason is twofold. First, in variational quantum AdaBoost, weak classifiers can be boosted to obtain a strong classifier as long as the weak classifiers are slightly better than random guess. Noise may degrade the weak classifiers, however, as long as they are still better than random guess, they can be boosted to obtain a strong classifier. Second, as PQCs are shallow, quantum classifiers are weak, but also, the classifiers are less affected by noise due to shallow circuits.

## 4 CONCLUSION

In the current NISQ era, quantum machine learning usually involves a specification of a PQC and optimizing the trainable parameters in a classical fashion. Quantum machine learning has good generalization property while its trainability is generally poor. Ensemble methods are particularly appropriate to improve the trainability of quantum machine learning, and in turn help predict accurately. In this paper we theoretically establish the prediction guarantee of binary variational quantum AdaBoost, and numerically demonstrate that for multi-class classification problems, variational quantum AdaBoost not only can achieve high accuracy in prediction, but also help mitigate the impact of noise. For future work, it is interesting to incorporate ensemble methods to solve other practical tasks.

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

## A    TECHNICAL LEMMAS

In this section, we present several lemmas that will be used in our proof of Theorem 3.1.

We first introduce a lemma that can be used to bound the training error of binary quantum AdaBoost.

**Lemma A.1.** (Theorem 7.2, Mohri et al. (2018)) *The training error of the binary classifier returned by AdaBoost verifies:*

$$\widehat{R}_S\left(f\right) \leq \exp\left[-2\sum_{t=1}^{T}\left(\frac{1}{2} - \epsilon_t\right)^2\right]. \tag{11}$$

*Furthermore, if for all $t \in [T]$, $\epsilon_t \leq \frac{1}{2} - \gamma$, then*

$$\widehat{R}_S\left(f\right) \leq e^{-2\gamma^2 T}. \tag{12}$$

Then we recall a well-known result in machine learning which provides an upper bound on the generalization error.

**Lemma A.2.** (Theorem 3.3, Mohri et al. (2018)) *Let $\mathcal{G}$ be a family of functions mapping from $\mathcal{Z}$ to $[0,1]$. Then, for any $\delta > 0$, with probability at least $1 - \delta$ over the draw of an i.i.d. sample set $S = \{z_i\}_{i=1}^{n}$ according to an unknown distribution $\mathcal{D}$, the following inequality holds for all $g \in \mathcal{G}$:*

$$\mathbb{E}_{z \sim \mathcal{D}}\left[g\left(z\right)\right] \leq \frac{1}{n}\sum_{i=1}^{n} g\left(z_i\right) + 2\mathfrak{R}_n\left(\mathcal{G}\right) + \sqrt{\frac{\log\frac{1}{\delta}}{2n}},$$

*where $\mathfrak{R}_n\left(\mathcal{G}\right)$ denotes the expectation of the empirical Rademacher complexity $\widehat{\mathfrak{R}}_S\left(\mathcal{G}\right)$ of $\mathcal{G}$, defined as*

$$\mathfrak{R}_n\left(\mathcal{G}\right) = \mathbb{E}_{S \sim \mathcal{D}^n}\left[\widehat{\mathfrak{R}}_S\left(\mathcal{G}\right)\right] = \mathbb{E}_{S \sim \mathcal{D}^n}\mathbb{E}_{\boldsymbol{\sigma}}\left[\sup_{g \in \mathcal{G}}\frac{1}{n}\sum_{i=1}^{n}\sigma_i g\left(z_i\right)\right], \tag{13}$$

*where $\boldsymbol{\sigma} = \left(\sigma_1, \cdots, \sigma_n\right)^\top$, with $\sigma_i$s independent uniform random variables taking values in $\{-1, +1\}$.*

The following lemma relates the empirical Rademacher complexity of a new set of composite functions of a hypothesis in $\mathcal{H}$ and a Lipschitz function to the empirical Rademacher complexity of the hypothesis set $\mathcal{H}$.

**Lemma A.3.** *(Lemma 5.7, Mohri et al. (2018))* Let $\Phi_1, \cdots, \Phi_n$ be $l$-Lipschitz functions from $\mathbb{R}$ to $\mathbb{R}$ and $\boldsymbol{\sigma} = \left(\sigma_1, \cdots, \sigma_n\right)^\top$ with $\sigma_i$s independent uniform random variables taking values in $\{-1, +1\}$. Then, for any hypothesis set $\mathcal{H}$ of real-valued functions, the following inequality holds:

$$\frac{1}{n}\mathbb{E}_{\boldsymbol{\sigma}}\left[\sup_{h \in \mathcal{H}}\sum_{i=1}^{n}\sigma_i\left(\Phi_i \circ h\right)\left(x_i\right)\right] \leq \frac{l}{n}\mathbb{E}_{\boldsymbol{\sigma}}\left[\sup_{h \in \mathcal{H}}\sum_{i=1}^{n}\sigma_i h\left(x_i\right)\right].$$

In our work, the hypothesis set $\mathcal{H} = \left\{h_{\boldsymbol{\theta}} : \boldsymbol{\theta} \in [0, 2\pi]^K\right\}$, which is composed of PQC-based hypothesis $h_{\boldsymbol{\theta}}$ defined in the form of Eq. (7). We provide an upper bound of its Rademacher complexity in the following lemma (see Appendix C for a detailed proof).

**Lemma A.4.** *For the quantum hypothesis set $\mathcal{H} = \left\{h_{\boldsymbol{\theta}} : \boldsymbol{\theta} \in [0, 2\pi]^K\right\}$ with $h_{\boldsymbol{\theta}}$ being defined as in Eq. (7), its Rademacher complexity can be upper bounded by a function of the number of independent trainable quantum gates $K$ and the sample size $n$:*

$$\mathfrak{R}_n\left(\mathcal{H}\right) \leq 6\sqrt{\frac{K\log 7K}{n}} + 2\sqrt{\frac{K}{n}}. \tag{14}$$

## B    PROOF OF THEOREM 3.1

*Proof.* From Eqs. (3) and (8), it is straightforward to verify that evaluating the prediction error of the binary quantum AdaBoost classifier returned by Algorithm 2 is equivalent to evaluating $R(f)$ with $f = \sum_{t=1}^{T} \alpha_t h_t$ without the sign function.

Note that the quantum hypothesis set $\mathcal{H} = \left\{ h_{\boldsymbol{\theta}} : \boldsymbol{\theta} \in [0, 2\pi]^K \right\}$ is composed of $h_{\boldsymbol{\theta}}$ being defined as in Eq. (7), with range belonging to $[-1, 1]$. Thus, in general, $f = \sum_{t=1}^{T} \alpha_t h_t$ does not belong to $\mathcal{H}$ or its convex hull which is defined as

$$\text{conv}(\mathcal{H}) = \left\{ \sum_{k=1}^{p} \mu_k h_k : p \geq 1, \mu_k \geq 0, h_k \in \mathcal{H}, \sum_{k=1}^{p} \mu_k \leq 1 \right\}. \tag{15}$$

However, we can consider the normalized version of $f$, denoted by

$$\bar{f} = \frac{f}{\|\alpha\|_1} = \frac{f}{\sum_{t=1}^{T} \alpha_t}, \tag{16}$$

which belongs to the convex hull of $\mathcal{H}$. Moreover, since $\text{sgn}(f) = \text{sgn}(\bar{f})$, from Eqs. (3) and (8), we have

$$R(f) = R(\bar{f}), \text{ and } \widehat{R}_S(f) = \widehat{R}_S(\bar{f}). \tag{17}$$

To bound $R(\bar{f})$, let

$$\mathcal{G} = \left\{ \mathbb{I}_{y\bar{f}(\boldsymbol{x}) \leq 0} : (\rho(\boldsymbol{x}), y) \in \mathcal{X} \times \mathcal{Y}, \bar{f} = \frac{\sum_{t=1}^{T} \alpha_t h_t}{\|\alpha\|_1}, h_t \in \mathcal{H}, t \in [T] \right\}. \tag{18}$$

Then according to Lemma A.2, it yields

$$R(\bar{f}) \leq \widehat{R}_S(\bar{f}) + 2\mathfrak{R}_n(\mathcal{G}) + \sqrt{\frac{\log \frac{1}{\delta}}{2n}}. \tag{19}$$

Since the zero-one loss function is 1-Lipschitz, from Lemma A.3, we have

$$\mathfrak{R}_n(\mathcal{G}) \leq \mathfrak{R}_n(\text{conv}(\mathcal{H})). \tag{20}$$

Moreover, according to Lemma 7.4 in Mohri et al. (2018),

$$\mathfrak{R}_n(\text{conv}(\mathcal{H})) = \mathfrak{R}_n(\mathcal{H}). \tag{21}$$

Now we can combine Eqs. (17), (19)-(21) to yield

$$R(f) \leq \widehat{R}_S(f) + 2\mathfrak{R}_n(\mathcal{H}) + \sqrt{\frac{\log \frac{1}{\delta}}{2n}}. \tag{22}$$

Thus, if for all $t \in [T]$, $\epsilon_t \leq \frac{1}{2} - \gamma$, we can derive Theorem 3.1 by leveraging Lemma A.1 and Lemma A.4 to further bound Eq. (22). $\square$

## C    PROOF OF LEMMA A.4

We first introduce the notion of covering number, which is a complexity measure that has been widely used in machine learning.

**Definition C.1.** (Covering nets and covering numbers Dudley (2014)) Let $(\mathcal{H}, d)$ be a metric space. Consider a subset $\mathcal{K} \subset \mathcal{H}$ and let $\epsilon > 0$. A subset $\mathcal{N} \subseteq \mathcal{K}$ is called an $\epsilon$-net of $\mathcal{K}$ if every point in $\mathcal{K}$ is within a distance $\epsilon$ of some point of $\mathcal{N}$, i.e.,

$$\forall x \in \mathcal{K}, \exists y \in \mathcal{N} : d(x, y) \leq \epsilon.$$

The smallest possible cardinality of an $\epsilon$-net of $\mathcal{K}$ is called the covering number of $\mathcal{K}$, and is denoted by $N(\mathcal{K}, \epsilon, d)$.

For example, for Euclidean space $\left(\mathbb{R}^K, \|\cdot\|_\infty\right)$, the covering number $N\left([-\pi, \pi]^K, \epsilon, \|\cdot\|_\infty\right)$ is equal to $\lceil \pi/\epsilon \rceil^K$, where $\lceil \cdot \rceil$ denotes the rounding up function. Intuitively, the hypercube $[-\pi, \pi]^K$ can be covered by a number of $\lceil \pi/\epsilon \rceil^K$ $K$-dimensional hypercubes whose sides have the same length $2\epsilon$.

Then we introduce several technical lemmas that will be used in the proof of Lemma A.4.

The following lemma relates the distance between two unitary operators measured by the spectral norm to the distance between their corresponding unitary channels measured by the diamond norm.

**Lemma C.2.** *(Lemma 4, Supplementary Information for Caro et al. (2022)) Let $\mathcal{U}(\rho) = U\rho U^\dagger$ and $\mathcal{V}(\rho) = V\rho V^\dagger$ be unitary channels. Then,*

$$\frac{1}{2}\|\mathcal{U} - \mathcal{V}\|_\diamond \leq \|U - V\|.$$

*Here, $\|\cdot\|$ denotes the spectral norm, and the diamond norm of a quantum unitary channel $\mathcal{U}$ is defined as*

$$\|\mathcal{U}\|_\diamond = \max_{|\psi\rangle\langle\psi|} \|\mathcal{U}\left(|\psi\rangle\langle\psi|\right)\|_1,$$

*with $\|A\|_1 = \mathrm{Tr}\left[\sqrt{A^\dagger A}\right]$ being the trace norm.*

The following lemma translates the distance between $J$-qubit rotational operators to the distance of their corresponding angles.

**Lemma C.3.** *(Distance between rotational operators) Given an arbitrary $J$-qubit Pauli tensor product $P \in \{I, X, Y, Z\}^{\otimes J}$ and two arbitrary angles $\theta, \tilde{\theta} \in [0, 2\pi]$, the corresponding $J$-qubit rotational operators are $R(\theta) = e^{-i\frac{\theta}{2}P}$ and $R\left(\tilde{\theta}\right) = e^{-i\frac{\tilde{\theta}}{2}P}$, respectively. Then, the distance between the two operators measured by the spectral norm can be upper bounded as*

$$\left\|R(\theta) - R\left(\tilde{\theta}\right)\right\| \leq \frac{1}{2}|\theta - \tilde{\theta}|.$$

*Proof.* According to the definition of rotational operators, we have

$$R(\theta) - R\left(\tilde{\theta}\right) = \left(\cos\frac{\theta}{2} - \cos\frac{\tilde{\theta}}{2}\right) I^{\otimes J} - i\left(\sin\frac{\theta}{2} - \sin\frac{\tilde{\theta}}{2}\right) P,$$

whose singular value is $2\left|\sin\frac{\theta-\tilde{\theta}}{4}\right|$ with $2^J$ multiplicity. Thus,

$$\left\|R(\theta) - R\left(\tilde{\theta}\right)\right\| = 2\left|\sin\frac{\theta-\tilde{\theta}}{4}\right| \leq \frac{1}{2}|\theta - \tilde{\theta}|.$$

$\square$

In the main text, we only consider the ideally unitary channel for simplicity, namely, the PQC-based hypothesis is in the form of $h_{\boldsymbol{\theta}}(\boldsymbol{x}) = \mathrm{Tr}\left[ZU(\boldsymbol{\theta})\rho(\boldsymbol{x})U^\dagger(\boldsymbol{\theta})\right]$. To describe the noise effect, a general quantum channel $\mathcal{A}$ is defined by a linear map $\mathcal{A} : \mathcal{L}(\mathcal{H}_A) \to \mathcal{L}(\mathcal{H}_B)$, which is completely positive and trace preserving (CPTP). For a quantum channel $\mathcal{A}$, the diamond norm is defined as

$$\|\mathcal{A}\|_\diamond = \sup_{\rho \in \mathcal{D}(\mathcal{H}_{RA})} \|(\mathcal{I}_R \otimes \mathcal{A})(\rho)\|_1,$$

where $\mathcal{D}(\mathcal{H}_{RA})$ denotes the set of density operators acting on the Hilbert space $\mathcal{H}_{RA} = \mathcal{H}_R \otimes \mathcal{H}_A$, and $\mathcal{I}_R$ is the identity map on the reference system $\mathcal{H}_R$, whose dimension can be arbitrary as long as the operator $\mathcal{I}_R \otimes \mathcal{A}$ is positive semi-definite.

The following lemma can help generalize the results of Lemma A.4 and Theorem 3.1 from the case of unitary quantum channels described in the main text to those of noisy quantum channels. This means that the hypothesis functions can be generalized to $h_{\boldsymbol{\theta}}(\boldsymbol{x}) = \mathrm{Tr}\left[Z\mathcal{E}_{\boldsymbol{\theta}}(\rho(\boldsymbol{x}))\right]$, where the noisy channel $\mathcal{E}_{\boldsymbol{\theta}}$ is composed of $K$ trainable multi-qubit rotational gates and arbitrarily many non-trainable gates. Notice that the general case can be reduced to the ideally unitary case by letting $\mathcal{E}_{\boldsymbol{\theta}}(\rho) = U(\boldsymbol{\theta})\rho U^\dagger(\boldsymbol{\theta})$.

**Lemma C.4.** (Subadditivity of diamond distance; Proposition 3.48, Watrous (2018)) *For any quantum channels $\mathcal{A}$, $\mathcal{B}$, $\mathcal{C}$, $\mathcal{D}$, where $\mathcal{B}$ and $\mathcal{D}$ map from $n$-qubit to $m$-qubit systems and $\mathcal{A}$ and $\mathcal{C}$ map from $m$-qubit to $k$-qubit systems, we have*

$$\|\mathcal{AB} - \mathcal{CD}\|_\diamond \leq \|\mathcal{A} - \mathcal{C}\|_\diamond + \|\mathcal{B} - \mathcal{D}\|_\diamond.$$

The following lemma enables us to employ the covering number of one metric space to bound the covering number of another metric space.

**Lemma C.5.** (Covering numbers of two metric spaces; Lemma 3, Barthel & Lu (2018)) *Let $(\mathcal{H}_1, d_1)$ and $(\mathcal{H}_2, d_2)$ be two metric spaces and $f : \mathcal{H}_1 \to \mathcal{H}_2$ be bi-Lipschitz such that*

$$d_2\big(f(x), f(y)\big) \leq K d_1(x, y), \forall\, x, y \in \mathcal{H}_1,$$

*where $K$ is a constant. Then their covering numbers obey the following inequality as*

$$N(\mathcal{H}_2, \epsilon, d_2) \leq N(\mathcal{H}_1, \epsilon/K, d_1).$$

According to the above lemmas, we can derive the covering number of general noisy quantum models.

**Lemma C.6.** (Covering number of noisy quantum models) *If each element of $\boldsymbol{\theta}$ is selected from $[-\pi, \pi]$, then the covering number of the set of quantum channels $\{\mathcal{E}_{\boldsymbol{\theta}}\}$, each of which is composed of $K$ trainable multi-qubit rotational gates and arbitrarily many non-trainable quantum channels, can be upper bounded as*

$$N(\mathcal{E}_{\boldsymbol{\theta}}, \epsilon, \|\cdot\|_\diamond) \leq \left\lceil \frac{\pi K}{\epsilon} \right\rceil^K.$$

*Proof.* From the structure of $\mathcal{E}_{\boldsymbol{\theta}}$, we have

$$\|\mathcal{E}_{\boldsymbol{\theta}} - \mathcal{E}_{\tilde{\boldsymbol{\theta}}}\|_\diamond \leq \sum_{k=1}^{K} \left\| \mathcal{R}_k(\theta_k) - \mathcal{R}_k(\tilde{\theta}_k) \right\|_\diamond \tag{23}$$

$$\leq 2 \sum_{k=1}^{K} \left\| R_k^{(i_k, j_k)}(\theta_k) - R_k^{(i_k, j_k)}(\tilde{\theta}_k) \right\| \tag{24}$$

$$\leq \sum_{k=1}^{K} |\theta_k - \tilde{\theta}_k| \tag{25}$$

$$= \|\boldsymbol{\theta} - \tilde{\boldsymbol{\theta}}\|_1 \tag{26}$$

$$\leq K \|\boldsymbol{\theta} - \tilde{\boldsymbol{\theta}}\|_\infty, \tag{27}$$

where $\mathcal{R}_k(\theta_k)$ denotes the quantum channel corresponding to the $J$-qubit rotational operator $R_k^{(i_k, j_k)}(\theta_k)$. Here, Eq. (23) is derived by repeatedly using Lemma C.4 to erase the non-trainable quantum channels, Eqs. (24) and (25) are obtained from Lemma C.2 and Lemma C.3, respectively, and Eq. (27) owes to the relation between the $l_1$-norm and $l_\infty$-norm for vectors in $\mathbb{R}^K$.

Thus, according to Lemma C.5, we have

$$N(\mathcal{E}_{\boldsymbol{\theta}}, \epsilon, \|\cdot\|_\diamond) \leq N\left(\boldsymbol{\theta}, \frac{\epsilon}{K}, \|\cdot\|_\infty\right) = \left\lceil \frac{\pi K}{\epsilon} \right\rceil^K.$$

$\square$

Now we prove Lemma A.4 by leveraging the technique of proving Theorem 6 in Supplementary Information of Caro et al. (2022).

*Proof.* To bound the Rademacher complexity

$$\mathfrak{R}_n(\mathcal{H}) = \mathop{\mathbb{E}}_{S \sim \mathcal{D}^n} \mathop{\mathbb{E}}_{\boldsymbol{\sigma}} \left[ \sup_{h_{\boldsymbol{\theta}} \in \mathcal{H}} \frac{1}{n} \sum_{i=1}^{n} \sigma_i h_{\boldsymbol{\theta}}(\boldsymbol{x}_i) \right],$$

the main idea is to use the chaining technique to bound the empirical Rademacher complexity

$$\mathbb{E}_{\boldsymbol{\sigma}} \left[ \sup_{h_{\boldsymbol{\theta}} \in \mathcal{H}} \frac{1}{n} \sum_{i=1}^{n} \sigma_i h_{\boldsymbol{\theta}} \left( \boldsymbol{x}_i \right) \right]$$

in terms of covering number.

First, with respect to the diamond norm, for each $j \in \mathbb{N}_0$, there exists an $2^{-j}$-covering net denoted by $\mathcal{N}_j$ for the set of quantum channels $\{\mathcal{E}_{\boldsymbol{\theta}}\}$, satisfying $N_j \leq \lceil 2^j \pi K \rceil^K$. To be specific, for each $j \in \mathbb{N}$ and every parameter setting $\boldsymbol{\theta}$, there exists a quantum operator $\mathcal{E}_{\boldsymbol{\theta},j} \in \mathcal{N}_j$ such that

$$\|\mathcal{E}_{\boldsymbol{\theta}} - \mathcal{E}_{\boldsymbol{\theta},j}\|_{\diamond} \leq \frac{1}{2^j}.$$

For $j = 0$, the 1-covering net of $\{\mathcal{E}_{\boldsymbol{\theta}}\}$ is $\mathcal{N}_0 = \{0\}$. Moreover, according to Lemma C.6, the covering number $N_j$ can be upper bounded by $\lceil 2^j \pi K \rceil^K$.

Then, for any $k \in \mathbb{N}$, we have

$$\mathcal{E}_{\boldsymbol{\theta}} = \mathcal{E}_{\boldsymbol{\theta}} - \mathcal{E}_{\boldsymbol{\theta},k} + \sum_{j=k}^{1} \mathcal{E}_{\boldsymbol{\theta},j} - \mathcal{E}_{\boldsymbol{\theta},j-1},$$

and

$$\mathbb{E}_{\boldsymbol{\sigma}} \left[ \sup_{h_{\boldsymbol{\theta}} \in \mathcal{H}} \frac{1}{n} \sum_{i=1}^{n} \sigma_i h_{\boldsymbol{\theta}} \left( \boldsymbol{x}_i \right) \right] = \mathbb{E}_{\boldsymbol{\sigma}} \left[ \sup_{h_{\boldsymbol{\theta}} \in \mathcal{H}} \frac{1}{n} \sum_{i=1}^{n} \sigma_i \mathrm{Tr} \left[ Z \mathcal{E}_{\boldsymbol{\theta}} \left( \rho \left( \boldsymbol{x}_i \right) \right) \right] \right]$$

$$\leq \frac{1}{2^k} + \frac{6}{\sqrt{n}} \sum_{j=1}^{k} \frac{1}{2^j} \sqrt{\log N_j} \tag{28}$$

$$\leq \frac{1}{2^k} + \frac{6}{\sqrt{n}} \sum_{j=1}^{k} \frac{1}{2^j} \sqrt{K \log \lceil 2^j \pi K \rceil} \tag{29}$$

$$\leq \frac{1}{2^k} + 12 \sqrt{\frac{K}{n}} \int_{\frac{1}{2^{(k+1)}}}^{\frac{1}{2}} \sqrt{\log \left\lceil \frac{\pi K}{\alpha} \right\rceil} \mathrm{d}\alpha, \tag{30}$$

$$\leq \frac{1}{2^k} + 12 \sqrt{\frac{K}{n}} \int_{\frac{1}{2^{(k+1)}}}^{\frac{1}{2}} \sqrt{\log \frac{7K}{2\alpha}} \mathrm{d}\alpha, \tag{31}$$

where Eq. (28) is derived following a similar analysis as that in Supplementary Information of Caro et al. (2022).

By taking the limit $k \to \infty$, we obtain

$$\mathbb{E}_{\boldsymbol{\sigma}} \left[ \sup_{h_{\boldsymbol{\theta}} \in \mathcal{H}} \frac{1}{n} \sum_{i=1}^{n} \sigma_i h_{\boldsymbol{\theta}} \left( \boldsymbol{x}_i \right) \right] \leq 12 \sqrt{\frac{K}{n}} \int_{0}^{\frac{1}{2}} \sqrt{\log \frac{7K}{2\alpha}} \mathrm{d}\alpha \tag{32}$$

$$= 12 \sqrt{\frac{K}{n}} \left[ \frac{\sqrt{\log 7K}}{2} + \frac{7\sqrt{\pi}K}{4} \mathrm{erfc} \left( \sqrt{\log 7K} \right) \right], \tag{33}$$

$$\leq 6 \sqrt{\frac{K \log 7K}{n}} + 2 \sqrt{\frac{K}{n}}, \tag{34}$$

where Eq. (33) is derived by using the integral

$$\int \sqrt{\log \frac{1}{\alpha}} \mathrm{d}\alpha = \alpha \sqrt{\log \frac{1}{\alpha}} - \frac{\sqrt{\pi}}{2} \mathrm{erf} \left( \sqrt{\log \frac{1}{\alpha}} \right), \tag{35}$$

with the error function defined by $\mathrm{erf}(x) = \frac{2}{\sqrt{\pi}} \int_0^x e^{-t^2} \mathrm{d}t$ and the complementary error function $\mathrm{erfc}(x) = 1 - \mathrm{erf}(x)$. Besides, we notice that the function $K \mathrm{erfc} \left( \sqrt{\log 7K} \right)$ decreases monotonically with $K \in \mathbb{N}$ and is upper bounded by $\mathrm{erfc} \left( \sqrt{\log 7} \right) \leq 0.0486$, so we have Eq. (34).

Thus, by taking expectations of both sides of Eq. (34) over the training sample set $S$, we have Eq. (14). $\qquad \square$

