# OpenReview forum: "Quantum AdaBoost with Supervised Learning Guarantee"
_ICLR.cc/2024/Conference — Submitted to ICLR 2024_

### Official Review · Reviewer_Vvq1 · 2023-10-29

**Soundness:** 2 fair
**Presentation:** 3 good
**Contribution:** 2 fair
**Rating:** 3
**Confidence:** 5

**Summary:**

This work exploits a quantum Adaboost method to enhance the generalization capability of quantum neural networks, where related theoretical analysis is provided. The work also demonstrates the ensemble architecture of quantum neural networks is promising on NISQ devices.

**Strengths:**

1. Using the Adaboost method for Quantum Neural Networks is interesting.

2. The theoretical bounds for the quantum Adaboost algorithm are a necessary contribution.

**Weaknesses:**

1. In section 2.1, using multi-qubit quantum gates for the parametric quantum circuits is not optimal, as real multi-qubit quantum gates have to suffer from more serious quantum noise and are harder to deal with the Barren Plateau problem. Why not use single quantum parametric gates?

2. The Eq. (5) associated with the analysis of empirical risk minimizer is incorrect for the quantum neural network (QNN). The output of  QNN relies on the measurement, resulting in an additional optimization bias that is related to the expectation of observables. The authors can refer to the Reference as:

Ref. Jun Qi, Chao-Han Huck Yang, Pin-Yu Chen*, Min-Hsiu Hsieh*, "Theoretical Error Performance Analysis for Variational Quantum Circuit Based Functional Regression," Nature Publishing Group, npj Quantum Information, Vol. 9, no. 4, 2023

3. Accordingly, the theoretical upper bound on Eq. (9) is not complete. An additional optimization bias corresponding to the optimization bias needs to be considered.

4. The quantum Adaboost algorithm in Algorithm 2 is basically identical to the classical Adaboost one. So, what are the quantum advantages of quantum neural networks against classical neural networks? Since the performance of the classical neural networks can be also boosted to better performance, it is not clear why the authors highlight the quantum Adaboost counterpart.

5. The authors do not provide a deeper discussion for the simulations as shown in Figure 4 and 5, why more Rounds are beneficial to the performance boost for the quantum Adaboost system? and why does the quantum Adboost method even attains worse results at the very beginning?

**Questions:**

1. Why more rounds T can be beneficial to the proposed Adaboost method?

2. Why not provide the classical neural networks to compare the Adaboost performance?

3. If using the same Adaboost algorithm, what are the quantum advantages of using quantum neural networks?

---

> ### Author Response · Authors · 2023-11-19
> **Response to Reviewer Vvq1 (Weaknesses)**
>
> Thank you very much for your insightful comments. We have thoroughly revised the manuscript to make our contributions clearer.
>
> We utilize multi-qubit quantum gates for the following reasons. 1) Parametric quantum circuits constructed by using only single quantum parametric gates are classical in essence and cannot provide sufficient expressibility for variational quantum algorithms to ensure a good approximation to the solution. 2) We agree with you that multi-qubit quantum gates may suffer from more noise and lead to Barren Plateau issues. There have been many schemes to mitigate the training issues. 3) In our experiments, we only employ 2-qubit quantum gates. 4) 2-qubit quantum gates can be realized in a native way to minimize the hardware noise. 5) 2-qubit quantum gates are necessary components in most of the existing parametric quantum circuits.
>
> Thank you for pointing out the reference. We have read it in detail. First of all, Eq. (5) in our paper is a widely-adopted figure of merit to depict the generalization error, see e.g.,
>
>   1)	Matthias C. Caro, et.al., Generalization in quantum machine learning from few training data, Nature Communications, 13:4919, 2022.
>   2)	Matthias C. Caro, et.al., Out-of-distribution generalization for learning quantum dynamics, Nature Communications, 14:3751, 2023.
>   3)	Hsin-Yuan Huang, et.al., Power of data in quantum machine learning, Nature Communications, 12:2631, 2021.
>
> In quantum machine learning, the prediction accuracy is what we really concern, but it cannot be directly accessed as both the label of unseen data and the distribution is unknown. Nevertheless, we do have access to the empirical or training error. The difference between the prediction error and the training error is referred to as the generalization error. It is unknown but we can bound it as we have done in our work.
>
> In your mentioned reference, they provided another way for error decomposition, but the so-called training error (whose meaning is different from ours) that results from the optimization bias of gradient-based algorithms is hard to bound. In fact, they did not give such a bound. In addition, to obtain their main results, they assumed a crucial PL condition which assumes that the norm of the cost function gradient is bounded below by the cost function multiplied by a parameter $\mu$, namely, $\frac{1}{2}\\|\nabla\mathcal{L}_S(\boldsymbol{\theta})\\|^2_2\geq\mu\mathcal{L}_S(\boldsymbol{\theta})$. This is a very strong assumption, as it is well-known that training quantum parameterized circuits often suffers from the Barren Plateau issue, where the cost gradient vanishes exponentially fast as the scale of the problem increases. Thus, the strong PL constraint may limit the applicability of their main result. Our work provides the first theoretical proof that for binary classification the prediction error of variational quantum AdaBoost can converge to 0 as the increase of the number of boosting rounds and sample size.

---

> > ### Comment · Reviewer_Vvq1 · 2023-11-23
> > **A follow-up response to the authors' feedback**
> >
> > I thank the authors' response to my comments on the paper. Although the authors tried to clarify the theoretical contribution to the quantum neural network, however, their results are not novel as AdaBoost is a commonly used method to aggregate weaker classifiers into a strong one, making the training error eventually lowered to 0. The AdaBoost method is not restricted to QNN, it can be used in all classical machine learning methods, and the authors do not demonstrate the quantum advantages over the classical ones in terms of generalization capability or efficient training process.
> >
> > As for the reference I recommended to the authors, the setup of the PL condition for QNN can indeed guarantee a low training error, and the PL condition can be relaxed by using a pre-training method that also ensures a low optimization bias, and the pre-trained model can be much more easily set up. The authors can refer to the following paper below:
> >
> > Jun Qi, et al., "Pre-Training Tensor-Train Networks Facilitate Machine Learning with Variational Quantum Circuits," arXiv:2306.03741v1.
> >
> > Overall, the theoretical contribution of this paper is limited, and I cannot raise my evaluation score.

---

> ### Author Response · Authors · 2023-11-19
> **Response to Reviewer Vvq1 (Questions)**
>
> - Why more rounds T can be beneficial to the proposed Adaboost method?
>
> The fact that more rounds are beneficial to the performance of quantum AdaBoost is owing to the characteristic of AdaBoost. Take Algorithm 2 for an illustration. When $t=1$, it starts from a selected base classifier whose error is less than 1/2. Then at each round ($t\in[T]$), a new base classifier, whose error is less than 1/2, is selected and the distribution is updated to have the effect of focusing more on the points incorrectly classified at the next round. After $T$ rounds, the classifier returned by AdaBoost is a non-negative linear combination of the base classifiers selected at each round. In Figure 4 and Figure 5, when $T=1$, namely, at the beginning stage of AdaBoost, the classifier is only a base classifier, which is weak. Thus, its performance is relatively bad. We have made this clearer in the current manuscript.
>
> - Why not provide the classical neural networks to compare the Adaboost performance?
> - If using the same Adaboost algorithm, what are the quantum advantages of using quantum neural networks?
>
> According to your suggestion, we have performed new experiments to compare our quantum AdaBoost with its classical counterpart. For classical AdaBoost, weak classifiers can be boosted to obtain a strong classifier which has a high level of training accuracy. However, it is generally hard to theoretically guarantee that the generalization error of the strong classifier is small. In contrast, for quantum machine learning, the generalization of quantum classifiers can be guaranteed, while the training is often difficult. In our paper we combine the advantages of quantum machine learning in generalization and classical ensemble methods in training to obtain our main result, namely, variational quantum AdaBoost with supervised learning guarantee for binary classification. In the revised manuscript, as demonstrated in Table 1, we numerically validate that while the training accuracy of quantum AdaBoost is lower than that of classical AdaBoost, the prediction accuracy on new data of quantum AdaBoost is better as compared with classical AdaBoost.
>
> ||QCNN+AdaBoost|QCNN+Bagging|CNN+AdaBoost|CNN+Bagging|
> |--|--|--|--|--|
> |Training Acc. |$0.975 \pm 0.002$| $0.898 \pm 0.006$|$0.980 \pm 0.004$| $0.982\pm0.004$|
> |Prediction Acc. |$\boldsymbol{0.973\pm0.001}$|$0.888\pm0.005$|$0.967\pm0.003$|$0.965\pm0.002$|
> |Base Classifier |$0.861\pm0.019$|$0.851\pm0.020$|$0.876\pm0.051$ |$0.872\pm0.045$|

---

### Official Review · Reviewer_Pgtb · 2023-10-30

**Soundness:** 2 fair
**Presentation:** 3 good
**Contribution:** 2 fair
**Rating:** 5
**Confidence:** 3

**Summary:**

This work mainly proposes a quantum counterpart of AdaBoost algorithm, giving the theoretical analysis of prediction error on the binary classification problem and numerically providing the proof-of-principle experiments.

**Strengths:**

The manuscript is well-written and clearly introduces the quantum AdaBoost under the framework of the variational quantum algorithm. From theoretical and numerical perspectives, it demonstrates the feasibility of improving the performance of a quantum learning model by combining a few weaker ones.

**Weaknesses:**

1. With the limitation of the system size and circuit depth, the study aims to combine a few weaker quantum classifiers to improve the performance. The manuscript does not show the quantum advantages of the proposed model compared to classical ones from neither theoretical nor numerical. For instance, giving some tasks that are challenging for classical algorithms but can be surpassed by quantum learning models.
2. The technical and conceptual contributions are not significant enough.
3. In numerics, it only provides a single run which is insufficient.

**Questions:**

1. The algorithm 2 points out the error of the base classifier should be small. However, under the limitation of circuit depth, how to guarantee the classifier $h_t$ has a small test error meanwhile with shallow circuit depth.
2. In theorem 3.1, it points out that the generalization error is bounded by the number of training samples $n$ and independent trainable gates $K$. Since we can increase the number of $K$ to improve the model and decrease the error, however, why do we increase $K$, the bound is getting worse, which is not reasonable.
3. The quantum advantages are not quite clear, is there any evidence that the proposed method gives quantum advantages?

---

> ### Author Response · Authors · 2023-11-19
> **Response to Reviewer Pgtb**
>
> Thank you very much for your insightful comments. We have thoroughly revised the manuscript to make our contributions clearer.
>
> Finding various practical tasks that can demonstrate quantum superiority remains an open question. Here, we combine the advantages of quantum machine learning in generalization and classical AdaBoost in training to provide the first theoretical proof that for binary classification the prediction error of variational quantum AdaBoost can converge to 0 as the increase of the number of boosting rounds and sample size. In the revised manuscript, we also numerically validate that while the training accuracy of quantum AdaBoost is lower than that of classical AdaBoost, the prediction accuracy on new data of quantum AdaBoost is better as compared with classical AdaBoost. We have demonstrated the new numerical results in Table 1 in the revised manuscript. And we have performed more runs of experiments and all the results support our conclusion.
>
> ||QCNN+AdaBoost|QCNN+Bagging|CNN+AdaBoost|CNN+Bagging|
> |--|--|--|--|--|
> |Training Acc. |$0.975 \pm 0.002$| $0.898 \pm 0.006$|$0.980 \pm 0.004$| $0.982\pm0.004$|
> |Prediction Acc. |$\boldsymbol{0.973\pm0.001}$|$0.888\pm0.005$|$0.967\pm0.003$|$0.965\pm0.002$|
> |Base Classifier |$0.861\pm0.019$|$0.851\pm0.020$|$0.876\pm0.051$ |$0.872\pm0.045$|
>
> - The algorithm 2 points out the error of the base classifier should be small. However, under the limitation of circuit depth, how to guarantee the classifier $h_t$ has a small test error meanwhile with shallow circuit depth.
>
> Algorithm 2 only requires that quantum weak classifiers are slightly better than random guess, which is easy to be satisfied. We have made this clearer in the revised version.
>
> -  In theorem 3.1, it points out that the generalization error is bounded by the number of training samples $n$ and independent trainable gates $K$. Since we can increase the number of $K$ to improve the model and decrease the error, however, why do we increase $K$, the bound is getting worse, which is not reasonable.
>
> When increasing $K$, the bound is getting worse. This is owing to the Occam’s Razor principle: *Plurality should not be posited without necessity*. Specifically, there is a trade-off between reducing the training error versus controlling the size of the hypothesis set: a larger hypothesis set ($K$ is larger) could help reduce the training error, but is penalized by the second and third terms in Eq. (9). For a similar training error, it suggests using a smaller hypothesis set.
>
> - The quantum advantages are not quite clear, is there any evidence that the proposed method gives quantum advantages?
>
> Finding various practical tasks that can demonstrate quantum superiority remains an open question. Here, we utilize the advantage of quantum machine learning in generalization to provide the first theoretical proof that for binary classification the prediction error of variational quantum AdaBoost can converge to 0 as the increase of the number of boosting rounds and sample size.

---

### Official Review · Reviewer_5yFC · 2023-11-09

**Soundness:** 2 fair
**Presentation:** 2 fair
**Contribution:** 2 fair
**Rating:** 5
**Confidence:** 3

**Summary:**

This paper considers quantum ensemble learning when the quantum classification models are used as weaker learners. The generalization error bound is given for this type of ensemble learning. The authors also give empirical evidence of improved accuracy by quantum ensemble learning.

**Strengths:**

*Quality*

- The analysis of ensemble learning when the weak learners are quantum models are provided with rigor.  This analysis follows the standard routine for ensemble learning, and is technically sound to the best of my knowledge.

- The experimental settings make sense to me.

*Clarity*

- This paper is in general well written.

**Weaknesses:**

*Novelty*

- Limited novelty in theoretical analysis. The proof seems a straightforward combination of the standard analysis for ensemble learning and well-established lemmas for quantum models. Thus, the novelty of the theoretical analysis in this paper is limited.

- Limited novelty in the findings. The finding that ensemble learning can improve upon weak learners is not new to most ML audience. Thus, the key findings in this paper are not novel.

*Significance*

- Due to the limited novelty, this work seems of limited significance in both theoretical machine learning and quantum machine learning.

*Reproductivity*

- As there is no code for this work, it is unclear whether the empirical results are reproductible.

**Questions:**

*Question 1: Theoretical novelty*

The analysis in this paper seems a direct combination of existing tools for ensemble learning and quantum classifiers. What are the non-trivial theoretical points in this work?

*Question 2: Difference with related work*

What are the main differences between this work and existing works for ensemble learning and quantum machine learning? Please respond precisely.

*Question 3: New findings*

What are the main differences in experiments between the quantum weak learners and weak (classical) classifiers?

---

> ### Author Response · Authors · 2023-11-19
> **Response to Reviewer 5yFC**
>
> Thank you very much for your insightful comments. We have thoroughly revised the manuscript to present our contributions and the differences from existing works more clearly.
>
> - Theoretical novelty
>
> To the best of our knowledge, our work provides the first theoretical proof that for binary classification the prediction error of variational quantum AdaBoost can converge to 0 as the increase of the number of boosting rounds and sample size. To be specific, for classical AdaBoost, weak classifiers can be boosted to obtain a strong classifier which has a high level of training accuracy. However, it is generally hard to theoretically guarantee that the generalization error of the strong classifier is small. In contrast, for variational quantum machine learning, the generalization of quantum classifiers can be guaranteed, while the training is often difficult. In this paper we combine the advantages of quantum machine learning in generalization and classical ensemble methods in training, and utilize their analytical tools to obtain our main result, namely, variational quantum AdaBoost with supervised learning guarantee for binary classification.
>
> - Difference with related work
>
> We have revised Sec. 1.2 Related Work to make it clearer. For your convenience, we list the differences between our work and existing works in the following.
>
>    1)	As compared with classical AdaBoost, the training accuracy of quantum AdaBoost is lower, since it is often harder to train a quantum classifier than to train a classical classifier. However, as the generalization is easy to be guaranteed for quantum classifiers, the prediction accuracy on new data of quantum AdaBoost is better as compared with classical AdaBoost. We have performed new experiments to demonstrate this in Table 1 in the revised manuscript.
>    2)	Various quantum versions of AdaBoost have been proposed, such as Arunachalam & Maity, 2020; Wang et al., 2021b; Ohno, 2022. In their works, they employed quantum subroutines, e.g., mean estimation and amplitude amplification, to update quantum weak classifiers and estimate the weighted errors. Therefore, the realizations of these quantum versions of AdaBoost are beyond the scope of current noisy intermediate-scale quantum (NISQ) devices. In contrast, in our work we utilize variational quantum classifiers realized on the current NISQ circuits, which are obtained through a quantum-classical hybrid way.
>    3)	Although the variational quantum AdaBoost has been investigated, e.g., in Li et al., 2023, its performance is validated only through numerical simulations. In our work, we theoretically guarantee the performance of quantum AdaBoost.
>    4)	In the current NISQ era, quantum machine leaning is generally weak owing to limited scale of quantum circuits and inevitable influences of noise. Thus it is better to utilize the ensemble methods, e.g., AdaBoost, to combine weak learners to obtain a strong leaner.
>
> - New findings
>
> The training accuracy of quantum weak learners is lower than that of classical ones, as it is often harder to train quantum learners than to train classical learners. In our new experiments (illustrated in Table 1), under the same number of optimization iterations, the average training accuracy of quantum AdaBoost (having 120 parameters) is 0.975 with standard deviation 0.002, while the classical AdaBoost (having 787 parameters) has a higher level of average training accuracy which is 0.980 with standard deviation 0.004. In contrast, as the generalization is easy to be guaranteed for quantum classifiers, the average prediction accuracy on new data of quantum AdaBoost is 0.973 with standard deviation 0.001, which is better than that of the classical AdaBoost being 0.967 with standard deviation 0.003. We have demonstrated these results in Table 1 in the revised manuscript.
>
> ||QCNN+AdaBoost|QCNN+Bagging|CNN+AdaBoost|CNN+Bagging|
> |--|--|--|--|--|
> |Training Acc. |$0.975 \pm 0.002$| $0.898 \pm 0.006$|$0.980 \pm 0.004$| $0.982\pm0.004$|
> |Prediction Acc. |$\boldsymbol{0.973\pm0.001}$|$0.888\pm0.005$|$0.967\pm0.003$|$0.965\pm0.002$|
> |Base Classifier |$0.861\pm0.019$|$0.851\pm0.020$|$0.876\pm0.051$ |$0.872\pm0.045$|
>
> - As there is no code for this work, it is unclear whether the empirical results are reproductible.
>
> We will release the codes when preparing the camera-ready version. If you are interested in reproducing our results, we are also ready to share these codes with you.

---

> > ### Comment · Reviewer_5yFC · 2023-11-22
> >
> > Thanks for the response. I do not have further questions.

---

### Official Review · Reviewer_9hGs · 2023-11-10

**Soundness:** 2 fair
**Presentation:** 3 good
**Contribution:** 1 poor
**Rating:** 3
**Confidence:** 3

**Summary:**

The paper proposes to apply AdaBoost to ensemble models of parameterized quantum circuits for binary or multi-class classification of quantum states. An additional variant of the AdaBoost algorithm is proposed which is tailored for binary classification. Theoretical bounds and numerical demonstrations are presented in the paper.

**Strengths:**

The paper is well-presented. The algorithms and theorems are easy to follow.

**Weaknesses:**

The proposed method, named quantum AdaBoost, does not exploit the underlying quantum information. Effectively, it treats the PQCs as arbitrary weak classifiers that can take quantum states as input, and use AdaBoost to ensemble them. The novelty of this work is not thoroughly justified by the evidence presented. Besides, it is confusing to claim that multiple quantum AdaBoost algorithms have already been proposed in the literature while naming the proposed framework as quantum AdaBoost. The comparison between these algorithms and frameworks is also not explained.

The main theorem of the paper seems to be a combination of previously known results, with an additional bound on the Rademacher complexity of PQCs. The ideas justifying its novelty are not illustrated.

The experiments have not considered other ensemble methods and only compare the proposed framework with the base models. The experiments where noises are present do not suffice to justify that the proposed method is robust to the noises.

**Questions:**

What are the differences between your quantum AdaBoost and other previously proposed quantum AdaBoost algorithms?

How does your framework compare to other ensemble methods for quantum classifiers?

Why does your framework mitigate the effects of noises in PQCs?

---

> ### Author Response · Authors · 2023-11-19
> **Response to Reviewer 9hGs**
>
> Thank you very much for your insightful comments.
>
> - What are the differences between your quantum AdaBoost and other previously proposed quantum AdaBoost algorithms?
>
> We have thoroughly revised the manuscript to more clearly present our contributions and the differences from other previously proposed algorithms. For example, we have changed the title to “Variational Quantum AdaBoost with Supervised Learning Guarantee” and rewritten Sec. 1.2 Related Work and Sec. 1.3 Our Contributions to make them clearer.
>
> To the best of our knowledge, our work provides the first theoretical proof that for binary classification the prediction error of variational quantum AdaBoost can converge to 0 as the increase of the number of boosting rounds and sample size. To be specific, for classical AdaBoost, weak classifiers can be boosted to obtain a strong classifier which has a high level of training accuracy. However, it is generally hard to theoretically guarantee that the generalization error of the strong classifier is small. In contrast, for variational quantum machine learning, the generalization of quantum classifiers can be guaranteed, while the training is often difficult. In this paper we combine the advantages of quantum machine learning in generalization and classical ensemble methods in training to obtain our main result, namely, variational quantum AdaBoost with supervised learning guarantee for binary classification.
>
> Various quantum versions of AdaBoost have been proposed, such as Arunachalam & Maity, 2020; Wang et al., 2021b; Ohno, 2022. In their works, they employed quantum subroutines, e.g., mean estimation and amplitude amplification, to update quantum weak classifiers and estimate the weighted errors. Therefore, the realizations of these quantum versions of AdaBoost are beyond the scope of current noisy intermediate-scale quantum (NISQ) devices. In contrast, in our work we utilize variational quantum classifiers realized on the current NISQ circuits, which are obtained through a quantum-classical hybrid way.
>
> - How does your framework compare to other ensemble methods for quantum classifiers?
>
> Following your suggestion, we have performed new experiments comparing our quantum AdaBoost with Quantum Bagging, classical neural networks+AdaBoost, and classical neural networks+Bagging. We find that our quantum AdaBoost has the best prediction accuracy on new data among all these results. The results are demonstrated in Table 1 in the revised manuscript.
>
> ||QCNN+AdaBoost|QCNN+Bagging|CNN+AdaBoost|CNN+Bagging|
> |--|--|--|--|--|
> |Training Acc. |$0.975 \pm 0.002$| $0.898 \pm 0.006$|$0.980 \pm 0.004$| $0.982\pm0.004$|
> |Prediction Acc. |$\boldsymbol{0.973\pm0.001}$|$0.888\pm0.005$|$0.967\pm0.003$|$0.965\pm0.002$|
> |Base Classifier |$0.861\pm0.019$|$0.851\pm0.020$|$0.876\pm0.051$ |$0.872\pm0.045$|
>
>
> - Why does your framework mitigate the effects of noises in PQCs?
>
> In addition to the depolarizing noise, we also perform new experiments under the amplitude damping noise and phase damping noise. All experimental results demonstrate that quantum AdaBoost can help mitigate the influence of these noises. The reasons are twofold. First, in our variational quantum AdaBoost, weak learners can be boosted to obtain a strong learner as long as the weak learners are slightly better than random guess. Noise may degrade the weak leaners, however, as long as they are still better than random guess, they can be boosted to obtain a strong learner. Second, as the quantum parameterized circuit is shallow, quantum learners are weak, but also, the classifiers are less affected by noise due to shallow circuits.

---

> > ### Comment · Reviewer_9hGs · 2023-11-21
> >
> > Thanks for the author's response.
> >
> > > However, it is generally hard to theoretically guarantee that the generalization error of the strong classifier is small.
> >
> > There exists well-established research on the theoretical guarantees of the generalization error of ensemble methods, which is essentially one of the motivations of ensemble methods. For example, check chapter 4 of [1].
> >
> > [1] Schapire R E, Freund Y. Boosting: Foundations and algorithms[J]. Kybernetes, 2013, 42(1): 164-166.

---

> > > ### Author Response · Authors · 2023-11-23
> > > **Response to Reviewer 9hGs**
> > >
> > > Thank you very much for your response and providing the reference.
> > >
> > > We acknowledge the well-established research on the generalization error in Adaboost. However, we would like to emphasize two significant distinctions between our work and the results presented in Chapter 4 of [1].
> > >
> > > Firstly, our results characterize the relationship between the generalization error in terms of the number of independent parameterized gates $K$ and the size of training dataset $n$. This can be easily checked, as opposed to relying on the abstract concept of VC dimension in [1]. It provides a direct guidance enabling us to estimate the order of the required data size for achieving a smaller generalization error.
> > >
> > > Secondly, in our results, the upper bound of the generalization error is independent of the number of rounds $T$,  whereas in [1] the upper bound of generalization error is $T$-dependent. As $T$ increases, the upper bound of the generalization error also increases. In contrast, in our results, as $T$ increases, the upper bound of the training error exponentially decreases, but the upper bound of the generalization error remains unchanged. So, our work provides the theoretical proof that for binary classification the prediction error of variational quantum AdaBoost can converge to 0 as the increase of the number of boosting rounds and sample size.
> > >
> > > [1] Schapire R E, Freund Y. Boosting: Foundations and algorithms[J]. Kybernetes, 2013, 42(1): 164-166.

---

### Author Response · Authors · 2023-11-21
**General response**

Dear reviewers and meta reviewers,

We appreciate your valuable comments and constructive suggestions, which really help improve the quality of this paper. We have thoroughly revised the paper with changes highlighted in a BLUE font.

Till now, it remains an open question to demonstrate quantum superiority in practical learning tasks, especially in current noisy intermediate-scale quantum devices. To this end, we combine the ensemble methods with currently implementable variational quantum circuits to outperform classical counterparts. On top of that, to the best of our knowledge, our work provides the first corresponding theoretical proof on performance guarantee for binary classification.

For your convenience, we briefly list the main changes as follows.

Firstly, we have highlighted the difference of our work from prior works including classical AdaBoost, existing quantum versions of AdaBoost (beyond the scope of current quantum devices), empirical results of quantum ensemble methods, and present our contributions more clearly, with adding “variational” to the title.

Secondly, we have added more experimental results including classical or noisy or ensembled controlled settings to show the remarkable performance of our efficient methods, which coincides with the theoretical guarantee.

Lastly, we have also improved the description of algorithms and addressed all technical questions in detail.

We kindly request you to inform us if there are any other concerns that might have been overlooked, or if you have new questions. Please feel free to put your further comments on OpenReview and we are always happy to answer any questions from you.

---

### Meta-Review · Area_Chair_o1zb · 2023-12-06

**Metareview:**

This work considers a quantum algorithm for boosting. It shows that the prediction error of variational quantum AdaBoost decreases with the number of boosting iterations and the sample size. It tests the performance of the algorithm on quantum convolutional neural networks.

**Strengths**
- The paper is well motivated and clearly written.
- Convergence analysis of the algorithm is provided.
- Numerical results are used to illustrate the practical performance of the approach.

**Weaknesses**
- The proposed method does not exploit the underlying quantum information. In fact, the PQCs are taken as arbitrary weak classifiers that can take quantum states as input, and AdaBoost is used to ensemble them. Therefore, the results of the paper are insignificant.
- Since the paper studies the standard AdaBoost algorithm, the convergence analysis almost follows the existing ones.

**Suggestions to authors**
- Authors are strongly suggested to think deeply about the problem by leveraging quantum information into the AdaBoost algorithm.

**Justification For Why Not Higher Score:**

The technical contribution is very limited.

**Justification For Why Not Lower Score:**

N/A

---

### Decision · Program_Chairs · 2024-01-16

Reject